# Accuracy of endoscopic staging and targeted biopsies for routine gastric intestinal metaplasia and gastric atrophy evaluation study protocol of a prospective, cohort study: the estimate study

Stella A V Nieuwenburg ,[1] William W Waddingham,[2,3] David Graham,[2] Manuel Rodriguez-Justo,[4] Katharina Biermann,[5] Ernst J Kuipers,[1] Matthew Banks,[2] Marnix Jansen,[2,4] Manon C W Spaander[1]

SAVN and WWW are joint first authors.

For numbered affiliations see end of article.

**Correspondence to**
Dr. Manon C W Spaander;
v.spaander@erasmusmc.nl

## ABSTRACT

**Introduction** Patients with chronic atrophic gastritis (CAG) and intestinal metaplasia (IM) are at risk of developing gastric adenocarcinoma. Their diagnosis and management currently rely on histopathological guidance after random endoscopic biopsy sampling (Sydney biopsy strategy). This approach has significant flaws such as under-diagnosis, poor reproducibility and poor correlation between endoscopy and histology. This prospective, international multicentre study aims to establish whether endoscopy-led risk stratification accurately and reproducibly predicts CAG and IM extent and disease stage.

**Methods and analysis** Patients with CAG and/or IM on standard white light endoscopy (WLE) will be prospectively identified and invited to undergo a second endoscopy performed by an expert endoscopist using enhanced endoscopic imaging techniques with virtual chromoendoscopy. Extent of CAG/IM will be endoscopically staged with enhanced imaging and compared with standard WLE. Histopathological risk stratification through targeted biopsies will be compared with endoscopic disease staging and to random biopsy staging on WLE as a reference. At least 234 patients are required to show a 10% difference in sensitivity and accuracy between enhanced imaging endoscopy-led staging and the current biopsy-led staging protocol of gastric atrophy with a power (beta) of 80% and a 0.05 probability of a type I error (alpha).

**Ethics and dissemination** The study was approved by the respective Institutional Review Boards (Netherlands: MEC-2018-078; UK: 19/LO/0089). The findings will be published in peer-reviewed journals and presented at scientific meetings.

**Trial registration number** NTR7661; Pre-results.

## INTRODUCTION

Gastric adenocarcinoma remains a major cause of cancer mortality and is the most commonly diagnosed malignant condition of the gastrointestinal (GI) tract.[1–4] Although incidence rates had previously been declining,

### Strengths and limitations of this study

► This is the first study to compare endoscopy-led risk stratification of premalignant gastric lesions using advanced imaging and targeted biopsies with white light endoscopy and random biopsies as a reference, performed at separate occasions with the endoscopist blinded to prior results.

► This study will additionally provide biobank biopsy and serum material for future biomarker analysis.

► A possible limitation of this study is that all procedures will be performed by expert endoscopists in teaching hospitals, therefore external reproducibility will be evaluated using interobserver variability of disease staging through video recordings.

► The same limitation holds for the histopathological evaluation of the biopsy samples for which a proportion of the samples will be reviewed and re-scored by a blinded second expert gastrointestinal histopathologist.

recent studies demonstrated this differs among population subgroups. For example, an increasing incidence of gastric adenocarcinoma among young white cohorts in Western countries was objectified. This may be due to an increasing prevalence of gastric cancer precursors among younger adults, in particular chronic atrophic gastritis (CAG), intestinal metaplasia (IM) and dysplasia.[5 6] These studies suggest that gastric cancer incidence rates may plateau or even increase again in the upcoming years. Importantly, with the exception of Japan and Korea, the majority of gastric cancers worldwide are diagnosed at later stage. This results in a poor prognosis with less than 30% 5year survival.[1 2 7] Japan's

earlier stage of diagnosis and superior 5 year survival highlight the need for earlier recognition and treatment.[8]

Endoscopic recognition of the premalignant stomach has long been problematic and limited by the ability of endoscopist and the imaging tools. A previous study demonstrated that 22% of high-grade dysplastic lesions and early gastric cancers were missed.[9 10] A meta-analysis and systematic review of endoscopy follow-up studies confirmed that a marked proportion of early gastric cancers are missed at endoscopy.[10] Therefore, current practice uses histology-based staging.[11 12] However, endoscopic imaging has significantly improved with high-definition endoscopes and imaging enhancement technologies now routinely available. Some recent studies already suggested that accurate endoscopic staging of CAG and gastric intestinal metaplasia (GIM) is achievable and can robustly predict gastric adenocarcinoma risk. Importantly, the interobserver and intraobserver reproducibility characteristics of endoscopic CAG and GIM severity assessment are in experienced hands moderate to excellent.[13–18] These marked improvements in endoscopic technology and the shift towards an endoscopy-led approach will empower the endoscopist to risk stratify individuals with greater accuracy and decrease the already huge burden placed on our endoscopy and histopathology departments. Therefore, the aim of this study is to evaluate if enhanced endoscopic imaging, including high-definition white light endoscopy (WLE) and virtual chromoendoscopy, alongside targeted biopsies, provides an accurate and reproducible assessment of CAG and IM disease extent and staging, when compared with the current practice of WLE and random biopsies through the Sydney protocol biopsy strategy.

## METHODS AND ANALYSIS
### Aims
The primary aim of this study is to assess the diagnostic accuracy for the endoscopic diagnosis of IM in Sydney biopsy locations comparing standard endoscopic staging with random biopsies with enhanced imaging with biopsies targeted to GIM.[19] The Standards for Reporting of Diagnostic Accuracy Studies (STARD) guidelines were followed.[20] Study sites are located in the Netherlands and the UK. Secondary objectives are to evaluate (a) reproducibility of endoscopic staging after expert review, (b) reproducibility of histopathology for detection of IM, (c) the number of dysplastic or neoplastic lesions detected and (d) effects of inspection time of gastric mucosa on diagnostic accuracy.

### Design
This is a prospective, multicentre registry study on the accuracy and reproducibility of enhanced endoscopic imaging, including high-definition WLE and virtual chromoendoscopy, for the staging of CAG and IM. Two upper endoscopies will be performed on two separate occasions (6–12 months in between) using standard white-light endoscopy plus random biopsies (current

diagnostic strategy) at the first endoscopy and enhanced endoscopic imaging with targeted biopsies (proposed diagnostic strategy) at the second endoscopy. We will compare both approaches using histopathology as a reference and assess the accuracy and reproducibility of enhanced endoscopic imaging (figure 1).

## PATIENT AND PUBLIC INVOLVEMENT
We maintained close links with patient alliances and interest groups, both in the Netherlands as well as in the UK. This close relationship informs our practice and is the basis for the current study design. We will engage closely with patient interest groups to communicate research findings and ensure that our deliverables are fit for purpose.

## PARTICIPANTS
### Sample size
For estimation of sample size, we assume that the diagnosis of CAG or IM on enhanced imaging and targeted biopsies must be set with at least a 90% sensitivity with WLE and random biopsies as a reference.[15] A power (beta) of 80% and a probability of type I error (alpha) of 0.05 will be handled. That purpose requires at least 234 patients to be recruited to show a 10% difference in sensitivity between enhanced endoscopy-led staging and the standard WLE.

### Recruitment
All patients (>18 years of age) referred to the endoscopy department for routine diagnostic upper GI endoscopy and diagnosed with CAG or IM between November 2018 and June 2020 are eligible for inclusion if able to give informed consent. Patients are excluded when having (1) gastric neoplasia not amenable to endoscopic resection, (2) no indication for Sydney biopsy staging on standard WLE, (3) significant comorbidity, (4) a coagulation disorder, (5) previous gastric surgery or (6) are unable to complete the biopsy protocol in either endoscopy session.

## INTERVENTIONS
### Baseline characteristics
All patients are asked to complete a questionnaire on lifestyle factors, medical history, past interventions, medication use and family history of gastric cancer.

### White light endoscopy
Patients referred to the endoscopy department for upper GI endoscopy for investigation of symptoms or for surveillance of a known condition will undergo their procedure on a standard diagnostic gastroscopy list. Patients found to have CAG or IM will be prospectively identified. During the initial procedure, patients will receive the current recommended practice. Current practice is to initially identify if gastric atrophy is present and to inspect the gastric mucosa for areas suspicious for dysplasia or malignancy. Any mucosal abnormalities suspicious for

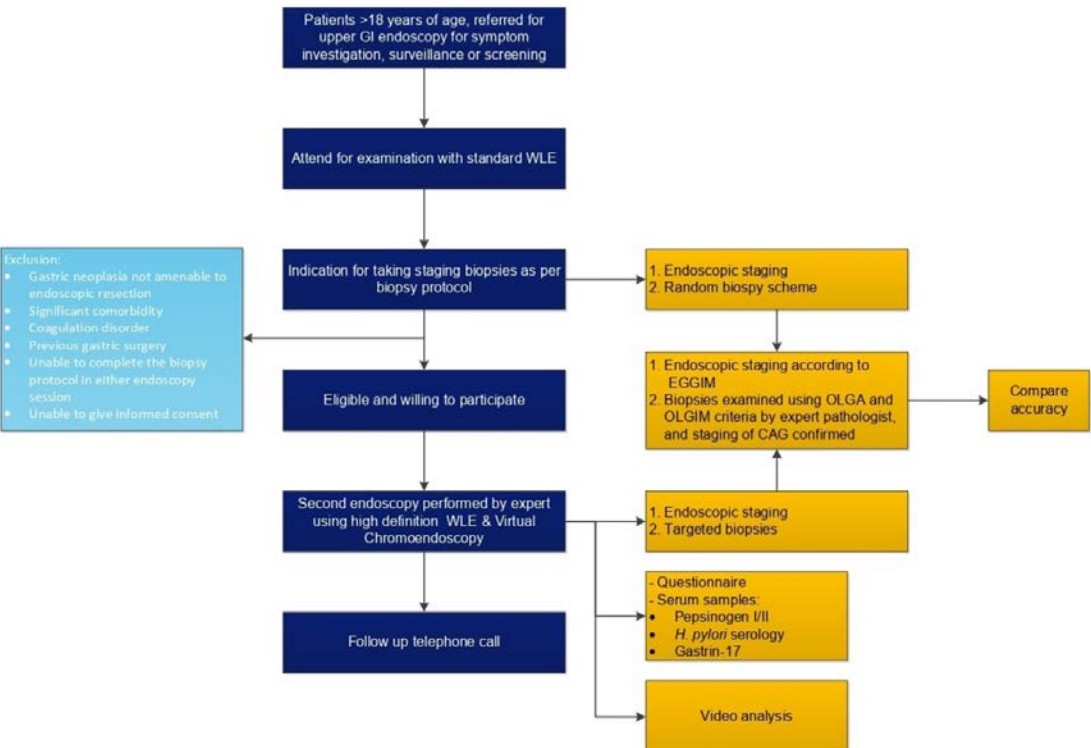

**Figure 1** Flowchart of the study design. CAG, chronic atrophic gastritis; EGGIM, endoscopic grading of gastric intestinal metaplasia; GI, gastrointestinal; *H. pylori, Helicobacter pylori*; OLGA, operative link for gastritis assessment; OLGIM, operative link for gastric intestinal metaplasia assessment; WLE, white light endoscopy.

dysplasia or malignancy are biopsied with tissue biopsies placed in separate containers. Following this, 10 random biopsies are taken according to the Sydney protocol (see also figure 2): 4 quadrant biopsies of the antrum, 2 biopsies from the incisura and 4 biopsies from the body of the stomach, respectively, 2 from the lesser curve, and 2 from the greater curve.

### Enhanced imaging endoscopy

Patients who opt to be recruited to the study will be invited for a second endoscopy at 6–12 months interval. This will be performed by one of the experts on this protocol. The endoscopists will be blinded to any previous endoscopy or biopsy results. This second endoscopy will be recorded

and performed using enhanced endoscopic imaging. Given that these patients will have recently undergone a complete upper GI endoscopy, while all anatomical landmarks will be viewed, the focus of this examination will be on the gastric mucosa. The endoscopist will record (1) the extent of gastric atrophy, (2) the presence and extent of IM in each of the aforementioned areas. This will be done using our simplified endoscopic metaplasia scoring system (GRAHAM Score) (table 1).

Biopsies will then be taken in the following manner: (1) areas of IM found in any of the Sydney protocol areas, (2) Sydney areas negative for GIM will be randomly biopsied, as control, to complete the assessment and (3) lesions suspicious for dysplasia or malignancy.

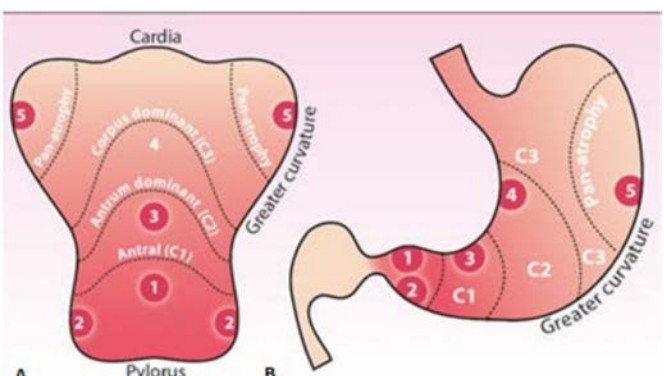

**Figure 2** Biopsy strategy. (A) Sydney protocol biopsy sites in the opened stomach along the greater curvature; (B) biopsy sites in the anatomical view.

| Table 1 Simplified endoscopic gastric intestinal metaplasia staging system: 'GRAHAM Score' | | |
|---|---|---|
| | **Focal/minimal metaplasia (<1/3 of surface coverage)** | **Moderate/ extensive metaplasia (>1/3 surface coverage)** |
| Antrum and incisura | 1 | 2 |
| Lesser curve | 1 | 2 |
| Greater curve | 1 | 2 |

## Histopathological assessment

Each biopsy will be reviewed at the teaching hospital by one of the expert GI histopathologists named on this protocol according to the established operative link for gastritis assessment and operative link for gastric intestinal metaplasia assessment staging systems.[21] Histopathologists will be blinded to whether biopsies were directed at areas suspicious for IM and to the biopsy results of WLE staging. A proportion of biopsy samples will be reviewed and rescored by a second expert GI histopathologist, who is blinded to the initial results. This is to ensure interobserver reproducibility for histopathological detection of IM.

## Serology assessment

A proportion of the collected serum will be used to assess *Helicobacter pylori* serology, pepsinogen I/II ratio and gastrin-17. The remaining serum will then be stored for use in future studies exploring the development of molecular biomarkers for gastric atrophy risk stratification.

## Data collection and management

All data collected for this study will be recorded in an anonymised format on a centralised, secure web-based platform (OpenClinica). Source data will be recorded in patients' notes or electronic health records, and hard copies of consent forms will be stored in a secure locked cabinet per site. All study data will be stored in a linked anonymised fashion against a study number, with the registry of study numbers stored separately on an encrypted database.

## Statistical analyses

For descriptive statistics, mean (±SD) will be used in case of a normal distribution of variables and median (25–75%) will be used for variables with a skewed distribution. Where appropriate, the Student's t-test or Mann–Whitney U test will be used.

Diagnostic accuracy of endoscopic diagnosis of CAG and IM is defined as the total number of directed biopsies that confirm the endoscopic impression of the presence or absence of IM divided by the total number of biopsies (accuracy=true positives+true negatives/all biopsies). Results will be compared with the histopathology outcomes using the $\chi^2$ test after multiple testing correction as well as kappa values for interobserver agreement among endoscopists and histopathologists.

After study completion, all videos will be collated and anonymised prior to expert panel review and estimation of the severity and extent of atrophy as well as IM. Five expert reviewers will review 50 videos each (kappa 0.4) for the purposes of assessing interobserver reliability. Sensitivity, specificity and global accuracy along with the 95% confidence intervals will be established. Duration of inspection time of gastric mucosa and its relation to diagnostic accuracy will be evaluated using, when appropriate, a paired t-test or Wilcoxon test. All tests will be two-sided.

## Ethics and dissemination

Results will be disseminated to potential users in academia and medical industries, through the standard routes of presentations, oral and posters, at local, national and international conferences, undergraduate and graduate teaching and through peer-reviewed publication. Efforts will be made to present work in a timely manner at key international meetings to encourage collaboration with research partners.

## DISCUSSION

The recently updated European MAnagement of Precancerous conditions and lesions in the Stomach (MAPS) guidelines recommend surveillance of patients with premalignant gastric mucosal lesions by performing endoscopy (preferably with advanced imaging) and taking random biopsies of the stomach for histopathological assessment. This enables the detection of progression to high-risk lesions and eventually cancer.[22] However, various studies indicate that a marked proportion of advanced gastric lesions are missed at a stage when these lesions are potentially still amenable to endoscopic management. This implies that the risk of undertreatment is undeniable.[9] The development of high-definition endoscopy and virtual chromoendoscopy has been a main focus of research in the past years and it has revolutionised the endoscopic assessment of the premalignant stomach by being superior to white light imaging.[23] The updated MAPS guidelines opt for the use of advanced imaging as the preferred surveillance method. Recently, Esposito *et al* showed a scoring tool based on endoscopic staging using Endoscopic Grading of Gastric Intestinal Metaplasia (EGGIM) with advanced imaging as a promising decision tool to identify patients at risk of gastric cancer.[24] However, currently there are no studies on how the use of advanced endoscopic imaging to detect IM of the stomach can be applied in countries with a low prevalence of IM. Still, histological confirmation is needed through random biopsies. Future steps are to evaluate the possible shift towards an endoscopy-led strategy now these marked improvements in endoscopic technology are within our reach. This prospective study was therefore designed to determine the validity of endoscopy-led staging of the premalignant stomach using advanced imaging and taking targeted biopsies for histological confirmation.

A previous comparative study between white light and high-definition endoscopy for the diagnosis of premalignant gastric lesions indeed showed a superior diagnostic accuracy of high-definition endoscopy.[15] However, one limitation was that WLE and high-definition endoscopy were performed during one occasion, which implied that the endoscopist was not blinded to the WLE results. Within the current protocol, we choose to perform the procedures on two separate occasions with blinding of the expert endoscopist to the previous WLE results.

Over the years, serological markers have shown major promise for predicting the presence and severity of gastric premalignant lesions.[25–27] Pepsinogens are serological markers for atrophy in the stomach and can be divided into pepsinogen I and II. A decreased PG I/II ratio indicates the presence of atrophic changes. Gastrin serum levels are indicative for gastric acid output and are increased in the

presence of atrophic changes.[27] The collection of serum samples was included in our protocol to strengthen risk stratification for progression of premalignant gastric lesions.

A few limitations of the study should be mentioned. All high-definition endoscopies will be performed by expert endoscopists at either site. A potential caveat with this design is the generalisability of the study outcomes to non-expert settings. To test this, we selected a panel of independent endoscopists who will review recorded endoscopy videos in order to assess interobserver variability. The same limitation holds for the histopathological evaluation of the biopsy samples. Therefore, a proportion of the samples will be reviewed and rescored by a blinded second expert GI histopathologist.

In conclusion, prospective validation of endoscopy-led staging of the premalignant stomach will provide the needed evidence for an endoscopy-led risk stratification of patients at risk for gastric adenocarcinoma. This will allow rational design of tiered screening and surveillance protocols to benefit early stage gastric cancer detection within at-risk populations. This will cause major implications for affected patients and general healthcare resource utilisation.

**Author affiliations**
[1]Gastroenterology & Hepatology, Erasmus MC University Medical Center, Rotterdam, The Netherlands
[2]Endoscopy, University College London Hospital, London, UK
[3]UCL Cancer Institute, University College London, London, UK
[4]Pathology, University College London Hospital, London, UK
[5]Pathology, Erasmus MC University Medical Center, Rotterdam, The Netherlands

**Correction notice** This article has been corrected since it was published. The licence has been updated.

**Contributors** EJK and MRB conceived the idea for the study, designed the protocol and supervise study execution. MCWS, DG and MJ supervise study execution. SAVN and WW drafted the manuscript. EJK, MRB, DG, MJ, MCWS, SAVN and WW analyse and interpret data. MRJ and KB interpret data. All authors provided critical revision of the manuscript for important intellectual content and approved the final draft of the protocol for submission.

**Funding** This work was supported by a grant from the Maag Lever Darm Stichting, Dutch Digestive Foundation grant number D17-22 and the Medical Research Council (clinical research training fellowship) number MR/S022244/1.

**Competing interests** None declared.

**Patient consent for publication** Not required.

**Ethics approval** The study was approved by the Institutional Review Boards of the Erasmus Medical Center (Erasmus MC) and University College London Hospitals (UCLH) (Netherlands: MEC-2018-078; UK: 19/LO/0089).

**Provenance and peer review** Not commissioned; externally peer reviewed.

**ORCID iD**
Stella A V Nieuwenburg http://orcid.org/0000-0001-6856-1814

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
