## [Reviewer comments · BMJ Open]

ARTICLE DETAILS

TITLE (PROVISIONAL)	ACCURACY OF ENDOSCOPIC STAGING AND TARGETED BIOPSIES FOR ROUTINE GASTRIC INTESTINAL METAPLASIA AND GASTRIC ATROPHY EVALUATION STUDY PROTOCOL OF A PROSPECTIVE, COHORT STUDY – THE ESTIMATE STUDY -
AUTHORS	Nieuwenburg, Stella; Waddingham, William; Graham, David; Rodriguez-Justo, Manuel; Biermann, Katharina; Kuipers, Ernst; Banks, Matthew; Jansen, Marnix; Spaander, Manon

VERSION 1 – REVIEW

REVIEWER	Bin Lv Department of Gastroenterology, First Affiliated Hospital of Zhejiang Chinese Medical University, 54 Youdian Road, Hangzhou 310006, China
REVIEW RETURNED	23-Jun-2019

GENERAL COMMENTS	This study aims to compare endoscopy-led risk stratification of premalignant gastric lesions using advanced imaging and targeted biopsies with white light endoscopy and random biopsies as a reference, compared to previous comparative study, this research choose to perform the procedures on two separate occasions with blinding of the endoscopist to the previous WLE results, which could avoid the influence of subject factors. There is one problem with the study design, whether it is appropriate to have a second endoscopy after an interval of 6 to 12-months ? A shorter interval of 3 to 6 months may be much more appropriate in case of pathological changes from gastric mucosa. Meanwhile, there are quite a lot of this sort of researches focus on this question recently, the creative and exploratory point of this study should be present clearly. In addition, gastric atrophy is an important premalignant condition, in this article, however, NBI endoscopic classification for the diagnosis of atrophy is not clear, and the diagnosis of atrophy amongst pathologists is not well elaborated too.
---

REVIEWER	Lahner Edith Sapienza University of Rome
REVIEW RETURNED	16-Jul-2019

GENERAL COMMENTS	I read with interest the study protocol and I have only a few comments.  - in the para regarding the sample size estimation the reference is lacking on which accuracy values are based - no mention is made about registration of medical or other interventions eventually occurring between the first (reference) gastroscopy and the second (index) gastroscopy. - the STARD guidelines are not reported in the reference list - the reference list is lacking at least one recent paper about EGGIM classification (Endoscopy 2019).
--

VERSION 1 – AUTHOR RESPONSE

> Reviewer(s)' Comments to Author:

>

> Reviewer: 1

>

> Reviewer Name

>

> Bin Lv

>

> Institution and Country

>

> Department of Gastroenterology, First Affiliated Hospital of Zhejiang

> Chinese Medical University, 54 Youdian Road, Hangzhou 310006, China

>

> Please state any competing interests or state 'None declared':

> None declared

> We have changed "all authors have nothing to declare" into "None declared".

> Please leave your comments for the authors below

This study aims to compare endoscopy-led risk stratification of premalignant gastric lesions using advanced imaging and targeted biopsies with white light endoscopy and random biopsies as a reference, compared to previous comparative study, this research choose to perform the procedures on two separate occasions with blinding of the endoscopist to the previous WLE results, which could avoid the influence of subject factors.

> There is one problem with the study design, whether it is appropriate to have a second endoscopy after an interval of 6 to 12-months ? A shorter interval of 3 to 6 months may be much more appropriate in case of pathological changes from gastric mucosa. Meanwhile, there are quite a lot of this sort of researches focus on this question recently, the creative and exploratory point of this study should be present clearly. In addition, gastric atrophy is an important premalignant condition, in this article, however, NBI endoscopic classification for the diagnosis of atrophy is not clear, and the diagnosis of atrophy amongst pathologists is not well elaborated too.

> We thank the reviewer for their valuable remarks. The second endoscopy, which will be performed 6-12 months after the first endoscopy, is for study purpose only and is not based on clinical grounds. Patients are informed about this and need to sign informed consent. The interval of 6-12 months was chosen to assure the endoscopists were blinded for the area that was previously sampled at the first endoscopy. In case pathological changes (low or high grade dysplasia, carcinoma) require more intensive follow up, participants will be in follow up according to the guideline (MAPS 2019). We have

added this in the text to avoid any unclarity.

We will take into account the presence of gastric atrophy. However, previous literature have showed that the use of OLGA is associated with a high inter- observer variability, therefore we decided not to use this classification in the current study but take note of the presence of gastric atrophy on both WLE and NBI endoscopy.

In the last few years studies have been performed evaluating the accuracy of NBI endoscopy for detection of gastric IM. Our study, however is the first study that compares WLE and NBI on two separate occasions, comparing standard endoscopic staging with random biopsies to enhanced imaging with biopsies targeted to GIM. Moreover, less is known how feasible this technique is in an area with a low prevalence of (extended) IM.

> Reviewer: 2

>

> Reviewer Name

>

> Lahner Edith

>

> Institution and Country

>

> Sapienza University of Rome

> Please state any competing interests or state 'None declared':

> None declared

> We have changed "all authors have nothing to declare" into "None declared".

> Please leave your comments for the authors below I read with interest

> the study protocol and I have only a few comments.

> - in the para regarding the sample size estimation the reference is

> lacking on which accuracy values are based

> We added the reference on which accuracy values are based.

> - no mention is made about registration of medical or other interventions eventually occurring between the first (reference) gastroscopy and the second (index) gastroscopy.

> We thank the reviewer for this important point. Since all patients are under clinical care within our hospitals chances are small that interventions will occur within the time span between reference and index endoscopy that we will not take notice of. We will register if this is the case, and through our questionnaire double check with our included patients about past interventions. We added "past interventions" explicitly in our protocol to clarify the above.

> - the STARD guidelines are not reported in the reference list

Thank you for this remark. We added the STARD guidelines in our methods and references.

> - the reference list is lacking at least one recent paper about EGGIM classification (Endoscopy 2019).

> We thank the reviewers for pointing out this reference. At the time of submission this paper was not published yet. We gladly added this reference and incorporated the results in our discussion.

VERSION 2 – REVIEW

REVIEWER	Bin Lv > Department of Gastroenterology, First Affiliated Hospital of Zhejiang > Chinese Medical University, 54 Youdian Road, Hangzhou 310006, China
REVIEW RETURNED	22-Aug-2019
GENERAL COMMENTS	The article has been modified to meet my requirements.
REVIEWER	Lahner Edith Sapienza University of Rome, Italy
REVIEW RETURNED	09-Aug-2019
GENERAL COMMENTS	No further comments.